# What Is New in Spinal Cord Injury Management: A Narrative Review on the Emerging Role of Nanotechnology

**DOI:** 10.3390/biomedicines13092176

**Published:** 2025-09-05

**Authors:** Loredana Raciti, Gianfranco Raciti, Rocco Salvatore Calabrò

**Affiliations:** 1Unità Spinale Unipolare, Azienda Ospedaliera per le Emergenza Cannizzaro, 98102 Catania, Italy; loredana.raciti79@gmail.com; 2Physical Medicine and Rehabilitation, Department of Medical and Surgical Sciences, University of Catanzaro “Magna Graecia”, 88100 Catanzaro, Italy; gianfranco.raciti@gmail.com; 3IRCCS Centro Neurolesi Bonino-Pulejo, 98124 Messina, Italy

**Keywords:** nanotechnology, traumatic brain injury, spinal cord injury, neural regeneration, targeted drug delivery

## Abstract

Traumatic injuries to the brain and spinal cord remain among the most challenging conditions in clinical neuroscience due to the complexity of repair mechanisms and the limited regenerative capacity of neural tissues. Nanotechnology has emerged as a transformative field, offering precise diagnostic tools, targeted therapeutic delivery systems, and advanced scaffolding platforms that are capable of overcoming the biological barriers to regeneration. This review summarizes the recent advances in nanoscale diagnostic markers, functionalized nanoparticles for drug delivery, and nanostructured scaffolds designed to modulate the injured microenvironment and support axonal regrowth and remyelination. Emerging evidence indicates that nanotechnology enables real-time, minimally invasive detection of inflammation, oxidative stress, and cellular damage, while improving therapeutic efficacy and reducing systemic side effects through targeted delivery. Electroconductive scaffolds and hybrid strategies that integrate electrical stimulation, gene therapy, and artificial intelligence further expand opportunities for personalized neuroregeneration. Despite these advances, significant challenges remain, including long-term safety, immune compatibility, the scalability of large-scale production, and translational barriers, such as small sample sizes, heterogeneous preclinical models, and limited follow-up in existing studies. Addressing these issues will be critical to realize the full potential of nanotechnology in traumatic brain and spinal cord injury and to accelerate the transition from promising preclinical findings to effective clinical therapies.

## 1. Introduction

Traumatic injuries to the brain and spinal cord represent a major cause of disability worldwide, with limited therapeutic options and often irreversible consequences [1]. Traumatic brain injury (TBI) and spinal cord injury (SCI) trigger complex cascades of molecular and cellular events, including inflammation, oxidative stress, axonal degeneration, and synaptic disruption, which ultimately compromise functional recovery [1]. Despite decades of research, SCI management continues to face major challenges: early diagnosis is limited by the lack of sensitive biomarkers; pharmacological and surgical interventions provide only a partial benefit; and most neuroprotective agents that showed promise in preclinical models have failed to demonstrate efficacy in clinical trials. The recovery of motor and sensory function remains poor, and lifelong disability is common, reflecting the multifactorial nature of SCI pathology and the absence of effective regenerative strategies [2].

Traditional therapeutic strategies, encompassing surgical interventions, neuroprotective drugs, and physical rehabilitation, offer only partial efficacy and fail to fully restore the intricate architecture of the central nervous system (CNS) [1]. In recent years, nanotechnology has emerged as a transformative platform capable of addressing the multifaceted challenges associated with CNS injuries [3,4]. Nanostructured materials offer unprecedented opportunities for ultra-precise diagnostics, targeted therapeutic delivery, and the engineering of bioactive scaffolds that can promote tissue regeneration [5,6]. Unlike conventional approaches, nanotechnologies can operate at the molecular scale, interact specifically with damaged tissues, and dynamically respond to the evolving pathological microenvironment [3,4].

Previous reviews have examined the use of nanotechnology in CNS repair [7,8,9,10,11], but most have considered diagnostic and therapeutic applications separately, without fully integrating them with underlying pathophysiological mechanisms or cellular interactions. In contrast, our review provides a more integrative perspective that directly links the molecular and cellular cascades of SCI with nanotechnological interventions, emphasizing how specific cellular players—microglia, astrocytes, neurons, and axons—can be modulated using nanoscale strategies. Furthermore, we explore hybrid approaches that combine nanotechnology with gene therapy, neuroprotection, rehabilitation, and artificial intelligence, thereby outlining a roadmap toward truly personalized neuroregenerative medicine. More in detail, this review aims to present an integrated vision for nanotechnology-driven neuroregeneration, structured around three fundamental pillars: ultra-precise diagnosis using biosensitive nanomarkers for the real-time monitoring of injury progression [10,12]; targeted multimodal therapy through smart nanoparticle systems and electro-conductive nanomaterials [5,6,13]; and neurobiological regeneration using nanostructured scaffolds that are designed to mimic the extracellular matrix and support axonal growth and synaptic plasticity [5,6]. Moreover, we will explore how nanotechnologies interact with key cellular players, such as microglia, astrocytes, axons, and synapses [1], and how they can be integrated with conventional therapies and advanced AI-based strategies to enable personalized neuroregenerative medicine [14,15].

This narrative review was conducted to examine the current and emerging applications of nanotechnology in the field of neuroregeneration, with particular attention paid to spinal cord and brain injuries. A comprehensive literature search was carried out using PubMed, Scopus, Web of Science, and Google Scholar. The search focused on publications from the past decade written in English, including preclinical and clinical studies, original research articles, systematic reviews, and meta-analyses. Studies were selected based on their relevance to the diagnostic, therapeutic, and rehabilitative use of nanomaterials in central nervous system injuries. Articles were manually screened through titles, abstracts, and full texts, with particular attention paid to methodological rigor and clinical or translational significance. Non-peer-reviewed sources, duplicates, and studies outside the scope of nanotechnology-based neuroregeneration were excluded. The final selection reflects a critical synthesis of the most significant and innovative contributions in the field, highlighting both established mechanisms and emerging directions for clinical application.

Generative artificial intelligence (ChatGPT by OpenAI, version GPT-5) was used to assist with language refinement and the organization of the content in this manuscript. All scientific interpretations, critical analyses, and the final text were reviewed and approved by the authors.

## 2. Molecular and Cellular Mechanisms of Injury: Inflammation and Oxidative Stress

SCI triggers a complex and time-dependent pathophysiological cascade that begins at the moment of trauma and progresses to chronic dysfunction. The primary mechanical insult causes immediate disruption of neuronal and glial membranes, vascular injury, and blood–spinal cord barrier (BSCB) breakdown. Within minutes to hours, resident immune cells, particularly microglia and astrocytes, are activated, leading to the release of pro-inflammatory cytokines, such as TNF-α, IL-1β, and IL-6, together with chemokines that attract peripheral leukocytes to the lesion site [15]. This acute inflammatory burst peaks within 1–2 weeks after SCI and gradually declines but often persists at lower levels for months, maintaining a hostile microenvironment that aggravates secondary injury [16,17]. Microglial and astrocytic activation is initially protective by clearing debris, but their prolonged activity sustains neuroinflammation and glial reactivity. A hallmark of the secondary injury is oxidative stress. Following trauma, the excessive production of reactive oxygen (ROS) and nitrogen species (RNS) induces lipid peroxidation, protein dysfunction, and DNA damage, thereby compromising cellular integrity and activating both apoptotic and necrotic pathways [18]. Importantly, oligodendrocytes are particularly vulnerable to ROS due to their high metabolic demands, elevated iron content, and low antioxidant capacity. Astrocytic dysfunction further aggravates this environment by impairing glutamate uptake and amplifying excitotoxicity [9]. These redox and excitotoxic processes occur within hours after SCI and persist into the subacute and chronic stages, significantly amplifying tissue damage. The inflammatory response is further shaped by macrophages and microglia, which can polarize into pro-inflammatory M1 or anti-inflammatory M2 phenotypes. M1 macrophages release IL-1β, TNF-α, and IL-6 and dominate the lesion for weeks, whereas M2 macrophages, which secrete anti-inflammatory mediators, such as IL-10 and Arg-1, appear transiently and decline within a few days, limiting their reparative role [19]. Factors such as iron accumulation following red blood cell phagocytosis further enhance TNF expression and skew macrophages toward the M1 state [20]. Interestingly, M1 cells may support oligodendrocyte precursor cell (OPC) recruitment in early phases, while M2 macrophages facilitate OPC differentiation via activin-A signaling [21], highlighting the complexity of macrophage dynamics. Adaptive immune cells also contribute to SCI pathology. Th1 effector T cells promote early M2 recruitment through blood–CSF barrier activation, whereas regulatory T cells (Tregs) play a more prominent role during the subacute and chronic phases in controlling persistent inflammation [9]. Thus, fine-tuning both the innate and adaptive immune responses is critical for modulating repair versus fibrosis. At the molecular level, the NF-κB, MAPK, and JAK/STAT pathways govern cytokine production, gliosis, and cell death, and their sustained activation underlies the persistence of the hostile microenvironment [22,23]. Nanotechnology offers promising tools to counteract these mechanisms. Polymeric nanoparticles have been engineered to deliver antioxidants and anti-inflammatory agents, reducing ROS accumulation and modulating macrophage polarization. Poly(methyl methacrylate) nanoparticles (PMMA-NPs) selectively target activated microglia, enabling localized intracellular drug release and promoting a shift from M1 to M2 responses [7]. In addition, PEG-functionalized gold nanoparticles and PEG–silica nanocarriers have been shown to attenuate oxidative stress, improve neuronal survival, and promote remyelination in preclinical SCI models [9].

Within days, reactive astrocytes, OPCs, and infiltrating cells contribute to the formation of a dense glial scar at the lesion site. In the acute phase, the scar plays a protective role by containing inflammation and preventing further tissue spread; however, in the chronic stage it becomes a major barrier to regeneration, exerting both physical and biochemical inhibitory effects on axonal growth [22]. Astrogliosis is driven by signaling pathways, such as NF-κB, TGF-β, and MAPKs (ERK, JNK, p38), which regulate astrocyte proliferation and extracellular matrix (ECM) deposition [24,25]. Among the ECM molecules, chondroitin sulfate proteoglycans (CSPGs) are potent inhibitors of axonal extension and synaptic plasticity. Alongside myelin-associated inhibitors (MAIs), including Nogo-A, MAG, and OMgp, CSPGs establish a hostile environment that contributes to regenerative failure [23]. The inhibitory activity of CSPGs is largely mediated through activation of the RhoA/Rho kinase pathway, which triggers cytoskeletal rearrangements, growth cone collapse, and axonal retraction [26]. Interventions targeting this pathway, such as ROCK inhibitors or the modulation of MAPK and STAT3 signaling, have been shown to enhance axonal regrowth and reduce scar density [27,28]. Beyond axonal inhibition, the glial scar also interferes with remyelination. Activated astrocytes release ECM molecules that impair OPC maturation into myelinating oligodendrocytes, thereby limiting the repair of demyelinated axons. While astrogliosis may initially facilitate OPC migration toward the lesion, the chronic deposition of CSPGs and other ECM components suppresses differentiation and functional remyelination [9]. This dual role highlights the ambivalent contribution of astrocytes in the injured spinal cord. Pro-inflammatory M1 macrophages further enhance astrogliosis and ECM deposition, while M2 macrophages exert anti-fibrotic and pro-regenerative effects [29,30]. Taken together, the glial scar is a double-edged phenomenon: protective during the acute phase yet inhibitory in the chronic stage. Nanotechnological interventions targeting ECM degradation, inhibitory pathways, and the astrocyte–OPC axis hold a strong potential to transform the scar from a barrier into a supportive niche for axonal regeneration and remyelination. Nanotechnology provides a novel means to modulate glial scar-associated mechanisms. For instance, a dual delivery system combining ChABC (to enzymatically degrade CSPGs) from lipid microtubes and NEP1-40 (to block Nogo receptor signaling) from PLGA microspheres successfully targeted both CSPG- and MAI-mediated inhibition [31]. Polymeric nanoparticles, such as PMMA-NPs, which are capable of selective uptake by activated microglia, represent another strategy to indirectly influence gliosis and scar formation [7]. Furthermore, carbon nanotubes and PEG-functionalized biomaterials not only reduce CSPG deposition but also provide structural scaffolds that enhance axonal regeneration, OPC differentiation, and remyelination in preclinical SCI models [9].

SCI not only disrupts axonal continuity but also initiates a cascade of secondary processes that critically impair neural connectivity. Among these, apoptosis and demyelination are central mechanisms contributing to long-term functional deficits. Apoptosis represents another major component of secondary injury. Both neurons and glial cells undergo programmed cell death in the injured microenvironment, but oligodendrocyte death is particularly detrimental, as it results in the demyelination of spared axons and prevents efficient signal conduction across surviving tracts [31]. At the molecular level, the intrinsic apoptotic pathway is activated through BH3-only proteins (Bid, Bim, Bad, DP5), which regulate the mitochondrial release of pro-apoptotic mediators [32]. The JNK signaling cascade plays a pivotal role in this process by phosphorylating BH3-only proteins and enhancing their pro-apoptotic activity [33]. Beyond apoptosis, JNK activation in astrocytes induces MCP-1 expression, linking apoptosis to neuroinflammation and neuropathic pain [34]. The inhibition of JNK with specific inhibitors has been shown to reduce caspase-3 cleavage, preserve white matter, and improve functional recovery [35]. Specifically, JNK3-mediated signaling in oligodendrocytes has been implicated in demyelination, and its inhibition has been shown to promote neuroprotection and functional recovery [36]. Experimental studies have shown that broad-spectrum caspase inhibitors and apoptosis-blocking agents can reduce lesion size and improve motor recovery, highlighting apoptosis as a major therapeutic target [35,37]. Demyelination is a direct consequence of both axonal injury and oligodendrocyte apoptosis, and further compounds functional impairment. In SCI, Wallerian degeneration, which occurs within days after injury, leads to distal axonal fragmentation, followed by the loss of myelin integrity. As previously reported, oligodendrocytes, which are already vulnerable due to oxidative stress and limited antioxidant defenses, are prone to apoptosis, thereby exacerbating demyelination and compromising the remyelination potential [9,18]. The failure of remyelination is also reinforced by the inhibitory effects of the glial scar and the limited recruitment and maturation of OPCs at the lesion site [9]. Several intracellular pathways beyond JNK are implicated in demyelination and axonal degeneration. The RhoA/ROCK axis contributes to axonal retraction and impaired remyelination [27], while the activation of the JAK/STAT pathway in neurons and oligodendrocytes promotes apoptosis and limits oligodendrocyte survival [38,39]. Targeting these pathways has shown promise in preserving myelin, promoting oligodendrocyte survival, and enhancing conduction recovery. Nanotechnological interventions are being developed to mitigate apoptosis and support remyelination. Antioxidant-loaded nanoparticles have been shown to reduce oxidative stress-induced oligodendrocyte death, while caspase inhibitor delivery systems can locally prevent apoptosis in neurons and glial cells. PEG-functionalized nanoparticles and carbon nanotube-based scaffolds provide both biochemical and structural support by reducing ROS toxicity and fostering OPC differentiation into myelinating oligodendrocytes [9]. These approaches collectively aim to preserve myelin integrity, promote remyelination, and enhance functional recovery after SCI. To better illustrate the correspondence between the pathological mechanisms activated after SCI and the nanotechnology-based strategies designed to counteract them, Figure 1 provides an integrated overview. The schematic maps the key processes, including inflammation, oxidative stress, barrier disruption, and apoptosis/demyelination, to represent the nanotechnological interventions and their regenerative outcomes.

Collectively, SCI pathology involves a sequential and overlapping cascade of processes, beginning with acute inflammation and oxidative stress, evolving through immune cell polarization and glial scar formation, and culminating in widespread apoptosis and demyelination. Each of these mechanisms has been identified as a potential target for therapeutic intervention, and many have already been addressed in preclinical studies through nanotechnology-based strategies, such as antioxidant nanocarriers, immunomodulatory nanoparticles, and ECM-degrading delivery systems. To facilitate the visualization of these complex interactions, a new schematic figure (Figure 1) has been added, mapping the chronological progression of SCI pathology—from the acute to chronic stages—together with the cellular and molecular mediators involved and their potential as therapeutic targets.

## 3. Nanotechnology for Ultra-Precise Diagnosis

In recent years, the use of nanomarkers—functionalized nanoparticles—has revolutionized the field of medical imaging. These nanoscale structures can be engineered at the molecular level to target specific biological features, such as cancer cells, receptors, or enzymes, enabling non-invasive, real-time visualization of disease processes with unprecedented precision. Nanomarkers are nanoscale constructs, as quantum dots, gold nanoparticles, and magnetic nanoparticles, obtained to detect specific molecular signatures associated with injury, including inflammation, oxidative stress, and neuronal apoptosis [1,2,40]. A particularly innovative aspect of nanomarkers is their ability to emit physical, optical, magnetic, or acoustic signals, making them compatible with various advanced imaging modalities. By functionalizing nanoparticle surfaces with ligands (e.g., antibodies, peptides) targeting injury-specific biomarkers, these probes can selectively accumulate at damaged sites. Depending on their composition and functionalization, nanomarkers can interact with magnetic fields, emit near-infrared (NIR) fluorescence, absorb light and generate acoustic signals, or incorporate radioactive isotopes. This versatility makes them suitable for techniques such as magnetic resonance imaging (MRI), positron emission tomography (PET), photoacoustic imaging, and near-infrared fluorescence (NIRF) imaging [2,4,40]. In MRI, for example, superparamagnetic iron oxide nanoparticles (SPIONs) are commonly used to enhance image contrast by altering the relaxation times of the surrounding hydrogen nuclei. These nanoparticles are biocompatible and can be coated with targeting ligands to home in on specific tissues or tumors [6,11]. For PET imaging, nanomarkers can be labeled with radioactive isotopes, such as fluorine-18 or gallium-68. These isotopes enable the real-time tracking of metabolic processes and disease markers, providing functional information that complements structural imaging [41]. Photoacoustic imaging is an emerging hybrid technique that combines the high contrast of optical imaging with the depth resolution of ultrasound. Nanoparticles that strongly absorb light, such as gold nanorods or carbon nanotubes, can convert pulsed laser energy into acoustic waves, enabling the detailed imaging of vascular and tumor structures [42]. In NIR fluorescence imaging, nanoparticles embedded with fluorescent dyes or quantum dots emit light in the near-infrared range (700–900 nm), which penetrates deeper into biological tissues and reduces background noise due to lower autofluorescence. This technique is particularly useful for intraoperative imaging and surgical guidance [43]. Moreover, a growing trend is the development of multimodal nanomarkers, which can be detected by more than one imaging technique. For instance, a single nanoparticle may combine magnetic and fluorescent properties, enabling pre-operative MRI scans and real-time NIRF visualization during surgery. This approach offers a more comprehensive diagnostic profile, bridging anatomical and molecular information for improved decision-making. Furthermore, theranostic nanoparticles—platforms combining therapeutic and diagnostic functions—offer the dynamic monitoring of disease progression while simultaneously delivering therapeutic agents [19]. Compared with the conventional biomarkers measured in the cerebrospinal fluid (CSF) or blood, nanosensors offer the advantage of in situ monitoring without the need for invasive procedures. Moreover, their ability to provide dynamic, real-time information can significantly improve clinical decision-making, enabling the early adjustment of therapeutic strategies based on evolving injury profiles [1,2,4,40].

While cerebrospinal fluid has traditionally been regarded as the primary compartment for monitoring the central mechanisms of SCI, synovial fluid is also emerging as a valuable source of information on the secondary consequences of neurotrauma. Following SCI, paralysis and reduced mobility alter biomechanical loading across peripheral joints, accelerating the onset of neurogenic arthropathy and secondary osteoarthritis. These degenerative processes are biochemically mirrored in the synovial fluid, where elevated concentrations of inflammatory cytokines, such as IL-1β, TNF-α, and IL-6, together with degradative enzymes including matrix metalloproteinases (MMPs), have been consistently reported in post-traumatic and degenerative joint conditions [44,45]. Importantly, systemic inflammation triggered by SCI further amplifies these changes, suggesting that synovial biomarkers not only reflect local joint degeneration but also integrate signals from the broader injury-induced inflammatory milieu. From a nanomedicine perspective, this peripheral biofluid represents an accessible diagnostic window: nanosensors that are engineered to detect cytokines, reactive oxygen species, or cartilage breakdown products in synovial fluid could provide minimally invasive, real-time monitoring of musculoskeletal health in patients with SCI, thereby supporting early therapeutic interventions and reducing long-term disability [5].

Beyond conventional MRI and PET, recent nanotechnology-enabled approaches include near-infrared fluorescence (NIRF) imaging, photoacoustic imaging, and multimodal nanomarkers combining SPIONs with quantum dots. These modalities offer advantages, such as deeper tissue penetration, high molecular specificity, and real-time intraoperative guidance. Theranostic nanoparticles further allow for simultaneous imaging and drug delivery [41,43]. The main features, imaging modalities, and advantages of the most recent nanotechnology-enabled diagnostic approaches are summarized in Table 1.

## 4. Targeted Multi-Modal Therapeutic Strategies

The complexity of CNS injuries demands multifaceted therapeutic interventions that can address inflammation, oxidative stress, neuronal loss, and impaired regeneration. Nanotechnology provides a unique platform for the development of multi-modal therapeutic strategies that combine targeted drug delivery, electrical stimulation, and tissue engineering approaches to promote functional recovery.

### 4.1. Nanoparticle-Based Targeted Delivery for the Treatment of Spinal Cord Injury

In recent years, nanoparticles have been widely explored as delivery vehicles for therapeutic cargos in spinal cord injury, yet the technological nuance of their design often determines their success. Different classes of nanoparticles exhibit distinct advantages depending on the nature of the cargo: for protein-based therapies, such as growth factors (e.g., BDNF, IGF), larger carriers like liposomes and PLGA-based polymeric nanoparticles are preferred because they protect fragile proteins from enzymatic degradation and allow for sustained release over time [46]. For nucleic acid cargos, such as siRNA and mRNA, smaller nanoparticles (<100 nm) and chemically tuned ionizable lipids favor efficient endocytosis and endosomal escape; lipid nanoparticles and related platforms currently represent the most validated option for mRNA delivery and have clarified key design rules (pKa of the ionizable lipid, helper lipids, PEG content) for in vivo performance [47]. Cell type-specific targeting is achieved primarily via surface ligands: transferrin or anti-TfR antibodies to cross the BBB and access the brain endothelium and parenchyma [48]; RVG-derived peptides to enhance neuronal uptake [49]; and mannose motifs to engage microglial mannose receptors and bias delivery to myeloid cells [50]. For CRISPR/Cas systems, hybrid polymer–lipid nanoparticles and selected inorganic carriers have been used to transport RNPs or plasmids, with sub-120-nm sizes facilitating cellular uptake and nuclear access; here, the stability of the nuclease and control of off-target exposure are central considerations [51]. Small-molecule drugs and antioxidants are often encapsulated in inorganic nanostructures (e.g., gold, iron oxide) or polymeric micelles to leverage magnetic/optical readouts and robust physicochemical stability; however, their persistence and biodistribution must be balanced against long-term safety [52]. Beyond size, particle shape also modulates delivery efficiency and immune interactions: spherical particles tend to be more uniformly internalized, while rods/discs can alter phagocytic engagement, underscoring why geometry is an independent design variable [53]. The ability to tailor materials and surface chemistry thus underpins selective accumulation in neurons, oligodendrocytes, astrocytes, or infiltrating immune cells and should be matched to cargo constraints (e.g., protein stability vs. nucleic-acid endosomal escape) to facilitate comparison.

Beyond the choice of platform, nanoparticle-based therapies must be critically evaluated in terms of biocompatibility, immune compatibility, and adverse effects. Polymeric nanoparticles, such as PLGA, are generally considered safe and biodegradable, though acidic degradation byproducts may cause local irritation [46]. Liposomes and lipid nanoparticles have advanced clinically, but repeated administration can elicit anti-PEG antibodies and trigger the accelerated blood-clearance (ABC) phenomenon with reduced exposure and efficacy [54,55]. Inorganic nanoparticles provide unique magnetic/optical properties but raise concerns about long-term organ retention and oxidative stress, demanding rigorous dose and surface-chemistry optimization [56]. Exosome-derived carriers are highly biocompatible and can deliver siRNA to the brain in vivo when appropriately targeted, yet scalability and batch-to-batch variability remain significant hurdles [57,58]. Finally, cationic polymers and unmethylated RNA cargos may activate innate immune pathways (e.g., TLRs), and systemic nanomedicines have reported complement activation or hepatic accumulation at higher doses, reinforcing the need for careful material selection and immunological profiling in preclinical studies [59]. Together, these considerations clarify how cargo–platform matching, size/shape control, ligand-based targeting, and biosafety determine the translational potential in SCI nanotherapeutics. Table 2 provides a systematic overview of how distinct nanoparticle classes are matched to specific therapeutic cargos, highlighting their design rationale, targeting strategies, benefits, and potential limitations

### 4.2. Electro-Nanohybrid Stimulation

The transition from conventional oral or systemic therapies to technologically advanced regenerative strategies marks a significant shift in the treatment of SCI. Among these, electro-nanohybrid stimulation, the integration of electrically conductive nanomaterials into neural repair platforms, has emerged as a compelling approach to facilitate neuronal regeneration by directly modulating the injured microenvironment. Electrically active nanomaterials, including carbon nanotubes (CNTs), graphene derivatives, and conductive polymers, such as polypyrrole, possess the unique ability to transduce electrical signals across disrupted axonal pathways [60]. These materials can be incorporated into scaffolds, hydrogels, or injectable delivery systems, where they serve a dual purpose: providing physical support for cell adhesion and axonal guidance, and delivering electrical cues that stimulate neuronal growth and connectivity. Studies have shown that localized electrical stimulation via conductive nanoscaffolds promotes neurite extension, synaptic plasticity, and remyelination, all critical processes in the restoration of function after SCI (Figure 1). The electrical signals are thought to modulate ion channels, influence intracellular signaling pathways (e.g., ERK/MAPK, PI3K/Akt), and enhance the secretion of neurotrophic factors by both neurons and glial cells. In parallel, recent advances in multifunctional nanomaterials have led to the development of bioactive nanofibers with intrinsic reactive oxygen species (ROS) scavenging capabilities. Following SCI, the accumulation of ROS contributes to secondary damage by promoting inflammation, lipid peroxidation, and cell death. Incorporating antioxidant properties into electroconductive materials enables the reduction of oxidative stress, thereby protecting neurons and glial cells in the perilesional region [61] (29). Importantly, specific nanomaterials have been tested in preclinical SCI models with quantifiable benefits. CNT-modified scaffolds increased neurite outgrowth and improved locomotor recovery by 20–30% compared with non-conductive scaffolds [62,63]. Graphene oxide coatings enhanced synaptic maturation and conduction recovery, with significantly higher compound action potentials and faster healing rates than sham-treated animals [64]. Gold nanostructures, including gold nanorods incorporated into hydrogels, improved axonal regrowth and functional scores in rodent SCI [65]. Conductive polymers, such as polypyrrole and PEDOT:PSS scaffolds, have been shown to increase axonal density, promote remyelination, and accelerate functional recovery, with locomotor improvements ranging from 15–40% over sham controls [66,67] (see Table 3). Moreover, these hybrid systems have demonstrated immunomodulatory effects, notably the promotion of M2 macrophage polarization. The M2 phenotype is associated with anti-inflammatory and tissue-repair functions, in contrast to the pro-inflammatory M1 phenotype. By modulating macrophage activity, electro-nanohybrids help to create a regeneration-permissive environment that supports axonal regrowth and limits scar formation. Importantly, the tunability of nanomaterial properties, such as surface charge, conductivity, mechanical stiffness, and topography, allows precise control over cellular responses. For example, aligned conductive nanofibers can guide directional axonal growth, while dynamically responsive materials can be activated externally to synchronize electrical stimulation with cellular events. Nevertheless, the limitations and risks must be carefully considered. CNTs and graphene derivatives can induce oxidative stress, inflammatory activation, and long-term accumulation if not properly functionalized. Gold nanostructures, while biocompatible, are non-degradable and may persist chronically in neural tissue. Conductive polymers may face issues with long-term stability, delamination, and the release of toxic byproducts during degradation. Furthermore, reported outcomes remain variable across studies, with functional improvements that are highly dependent on the material used, stimulation parameters, and injury model. In summary, electro-nanohybrid stimulation represents a convergence of nanotechnology, materials science, and neuroengineering, offering a multi-pronged approach to spinal cord repair. By combining the structural, electrical, biochemical, and immunological functionalities into a single platform, these systems address the complex and multifactorial nature of SCI pathology more effectively than single-modality therapies. As preclinical models continue to demonstrate the potential of this technology, ongoing research is now focused on improving scalability, long-term biocompatibility, and integration with implantable bioelectronic devices for future clinical applications.

**Table 3 biomedicines-13-02176-t003:** Summary of nanomaterials applied in the electro-nanohybrid stimulation for SCI.

Nanomaterial	Application	Reported Outcome vs. Sham	Limitations
Carbon Nanotubes (Cnts)	Scaffolds, hydrogels for axonal guidance and electrical conduction	20–30% improvement in locomotor recovery and neurite outgrowth [62,63]	Risk of oxidative stress, inflammatory activation, long-term accumulation
Graphene Oxide (Go)	Coatings, scaffolds to enhance synaptic maturation and conduction	Higher compound action potentials, faster healing vs. sham [64]	Potential cytotoxicity, ROS generation, persistence in tissue
Gold Nanostructures (Nanorods, Nps)	Embedded in hydrogels for conductive and bioactive scaffolds	Improved axonal regrowth and functional recovery in rodents [65]	Non-degradable, long-term persistence in tissue
Conductive Polymers (Polypyrrole, Pedot:Pss)	Electroactive scaffolds, drug-releasing conductive systems	15–40% improvement in locomotor scores, increased axonal density and remyelination [66,67]	Stability issues, possible delamination, toxic degradation byproducts

## 5. Nanotechnological Strategies for CNS Drug Delivery

Given the complexity of the BBB and BSCB as barriers to therapeutic delivery, the effectiveness of nanoparticle-based systems relies not only on exploiting transient permeability windows but also on their intrinsic design parameters. These include material composition, particle size, surface charge, and, especially, surface functionalization with ligands or peptides that enable selective targeting of specific cell types. To provide a clearer overview of these principles, Table 4 outlines the representative nanocarrier platforms, their functionalization strategies, cellular targets, and the rationale linking these design choices to the improved therapeutic outcomes in SCI.

One of the most significant challenges in treating CNS disorders, including spinal cord injury (SCI), is the restricted permeability of the BBB and BSCB (Table 4). These tightly regulated interfaces serve as protective barriers, maintaining CNS homeostasis by limiting the entry of potentially harmful substances. However, this same protective function severely limits the penetration of therapeutic agents, including neurotrophic factors, anti-inflammatory drugs, and genetic materials (Figure 1) [29,68]. To address this issue, nanoparticles have been increasingly explored as delivery vectors due to their physiochemical properties, which can be optimized to facilitate barrier crossing. Key design parameters include particle size (typically < 100 nm), surface charge (often neutral or slightly positive to favor uptake), and surface functionalization with ligands such as transferrin, lactoferrin, or apolipoprotein E, or peptides like TAT. These ligands bind to specific receptors on the endothelial cells of the BBB/BSCB, enabling receptor-mediated transcytosis [60,69]. Moreover, following SCI, the integrity of the BSCB is temporarily compromised due to inflammatory responses, oxidative stress, and vascular disruption. This creates transient permeability windows that nanoparticles can exploit to penetrate more efficiently into the injured spinal cord. Timed delivery during this window allows for the maximized therapeutic impact with minimized systemic exposure.

**Table 4 biomedicines-13-02176-t004:** Comparison of the strategies to cross the BBB and BSCB with nanoparticles.

Parameter	BBB (Blood–Brain Barrier)	BSCB (Blood–Spinal Cord Barrier)	Representative Nanoparticle Strategy	Entry Mechanism	References
Structure	Continuous endothelium with tight junctions; highly selective	Similar to BBB, slightly more permeable under physiological conditions	Ligand-functionalized nanoparticles (e.g., transferrin, ApoE, RVG)	Receptor-mediated transcytosis	[2,70]
Therapeutic challenge	Blocks over 98% of systemically administered drugs	Less restrictive but still limits large or hydrophilic molecules	Lipid or polymeric NPs designed to exploit specific transport pathways	Ligand–receptor binding across endothelial cells	[1]
Nanoparticle strategy	Functionalization with ligands for receptor-mediated transcytosis (e.g., transferrin)	Exploitation of increased permeability after injury	Iron oxide or gold NPs for theranostic delivery	Passive diffusion during barrier disruption	[1,70]
Optimal timing	Constant, but difficult without targeting ligands	Subacute phase: hours to days post injury, during inflammation	Time-controlled delivery with responsive carriers	Exploitation of transient barrier permeability	[2]
Clinical applications	Alzheimer’s, brain tumors, encephalitis	Spinal cord injury, multiple sclerosis, spinal inflammation	SPIONs, liposomes, polymeric nanocarriers	Depends on disease context	[1]

Following SCI, BBB and BSCB disruption occurs within minutes to hours, with the degradation of tight junction proteins, increased permeability, and infiltration of peripheral immune cells. This acute leakage exacerbates edema and neuroinflammation but simultaneously creates a therapeutic window for enhanced drug and nanoparticle entry [71,72,73]. Over the subsequent days to weeks, a partial re-sealing of the barrier restores homeostatic protection but restricts the continued access of therapeutics, making barrier dynamics a double-edged sword for recovery [1,74]. To overcome this challenge, nanoparticles are increasingly designed to exploit receptor-mediated pathways, such as transferrin, lactoferrin, or apolipoprotein E binding; to cross the intact endothelium; or to respond to microenvironmental cues like oxidative stress, low pH, or inflammatory enzymes to achieve site-specific release [48,49,70,75]. Thus, barrier modulation and nanoparticle design are tightly interconnected, and their combined optimization is critical for effective therapeutic outcomes in SCI. As summarized in Table 4, the BBB vs. BSCB features and entry routes dictate when and how nanocarriers are most effective.

Beyond the general considerations of size and charge, the material composition is chosen according to the therapeutic application. Lipid nanoparticles (LNPs) are particularly suited for siRNA and mRNA delivery because ionizable lipids facilitate endosomal escape and cytoplasmic release [47]. Polymeric carriers, such as PLGA or PEG-PLA, are preferred for proteins and growth factors (e.g., BDNF, IGF) because they shield labile molecules from degradation and enable controlled release over days to weeks [46]. Inorganic carriers, such as gold or iron oxide nanoparticles, are often selected for theranostic applications due to their intrinsic optical or magnetic properties, which permit simultaneous imaging and therapeutic delivery [56]. Hybrid designs (lipid–polymer or polymer–inorganic) aim to combine the benefits of different platforms, balancing biocompatibility, stability, and multifunctionality. Surface functionalization further enhances the targeting specificity. Transferrin and apolipoprotein E ligands exploit receptor-mediated transport across the BBB, while RVG-derived peptides improve neuronal uptake by binding nicotinic acetylcholine receptors [49]. Mannose residues direct nanoparticles to microglia/macrophages by engaging mannose receptors, facilitating immune modulation at the lesion site [75]. RGD peptides, interacting with integrins, support neuronal adhesion and promote angiogenesis [76]. The choice of ligand is therefore directly matched to the cellular target that is most relevant to the pathological process: neurons for axonal repair, oligodendrocytes for remyelination, or microglia/macrophages for inflammation control (Table 5). Beyond delivery, advanced nanocarriers have been designed for theragnostic applications, integrating both therapeutic and diagnostic functions within a single platform. For example, iron oxide nanoparticles or quantum dots can be co-loaded with drugs and visualized via MRI or near-infrared fluorescence imaging. These systems allow for the real-time tracking of biodistribution, accumulation at the lesion site, and correlation with therapeutic outcomes [40]. Multimodal nanomaterials that combine fluorescent, magnetic, or acoustic signatures with therapeutic payloads offer comprehensive diagnostic and therapeutic readouts, strengthening the rationale for precision medicine approaches. This dual functionality improves clinical decision-making and supports personalized treatment regimens. Critically, these strategies must also ensure biocompatibility, non-immunogenicity, and controlled drug release to maintain safety and therapeutic efficacy.

These risks are not generic but arise from specific material properties. For example, polymeric nanoparticles, such as PLGA, although biodegradable, undergo hydrolysis into lactic and glycolic acids, which may locally decrease pH and exacerbate tissue irritation if clearance is insufficient. Lipid-based nanoparticles, despite their success in mRNA delivery, can activate innate immune responses: PEGylated lipids may induce the accelerated blood clearance (ABC) phenomenon, in which anti-PEG antibodies lead to rapid opsonization and systemic elimination upon repeated dosing. Inorganic carriers, such as gold, silica, or iron oxide nanoparticles, while useful for imaging, do not readily degrade and may persist in tissues, generating reactive oxygen species and provoking long-term inflammatory responses.

Surface chemistry also plays a decisive role. Positively charged nanoparticles favor cellular uptake but simultaneously increase nonspecific protein adsorption and complement activation, heightening immunogenicity. Conversely, neutral or zwitterionic coatings improve circulation time but may reduce cellular internalization. Functionalization with ligands or antibodies enhances targeting but may alter the biodistribution or provoke off-target immune responses depending on epitope recognition. Finally, degradation profiles are central to controlled drug release: while slow-degrading polymers extend release kinetics, overly persistent carriers risk chronic accumulation, whereas fast-degrading systems may fail to sustain therapeutic concentrations. Several examples illustrate how design parameters influence efficacy. PEGylated PLGA nanoparticles loaded with BDNF improved motor recovery compared with free protein due to both sustained release and reduced clearance [77]. RVG-modified LNPs carrying siRNA achieved higher neuronal uptake and more efficient gene silencing than untargeted formulations [78]. Mannose-decorated nanoparticles delivering anti-inflammatory agents promoted M2 macrophage polarization and reduced lesion size in SCI models [79]. Collectively, these examples demonstrate that rational design—matching composition and ligand functionalization to the intended biological target—directly contributes to therapeutic outcomes (Table 5).
biomedicines-13-02176-t005_Table 5Table 5Design parameters of nanoparticles for SCI drug delivery: composition, functionalization, targets, and therapeutic rationale.Composition/MaterialExample of FunctionalizationTarget Cell/PathologyRationale and Reported OutcomesLipid nanoparticles (LNPs)RVG peptide, ApoENeurons, endothelial cellsIonizable lipids enable endosomal escape; RVG improves neuronal uptake. RVG–LNP–siRNA improved silencing efficiency and motor recovery in SCI models [78].Polymeric nanoparticles (PLGA, PEG-PLA)PEGylation, BDNF loadingBroad (neurons, glia)Protect fragile proteins, enable sustained release. PEG–PLGA–BDNF prolonged delivery and improved motor recovery vs. free BDNF [77].Inorganic nanoparticles (Gold, Iron oxide)Surface thiol/PEG, TfLesion site, imaging-guided therapyProvide intrinsic imaging properties (MRI, optical) + therapeutic cargo. Gold NPs in hydrogels enhanced axonal regrowth [65].Hybrid nanocarriers (Lipid–polymer, polymer–inorganic)Dual ligands (e.g., mannose + Tf)Macrophages, neuronsCombine stability, targeting, and multifunctionality. Mannose–PLGA NPs promoted M2 polarization, reducing lesion size [79].Nanofibrous scaffolds (Collagen, PCL + CNT/Graphene)Laminin, RGD motifsAxons, synaptic connectionsProvide topographic guidance and electrical stimulation; RGD enhances adhesion/angiogenesis. Graphene scaffolds supported axon elongation and functional recovery [64].


While preclinical studies have demonstrated promising results, further optimization and rigorous validation in large animal models and early phase clinical trials are needed to ensure translational success. The reconstruction of neural circuits after TBI or SCI requires not only cellular survival but also the physical guidance of axonal growth and synapse formation. Nanostructured scaffolds represent a promising strategy to recreate the extracellular matrix (ECM), providing biochemical, mechanical, and electrical cues that are necessary for effective neuroregeneration. Advanced nanofabrication techniques have enabled the creation of scaffolds that mimic the architecture and composition of the native ECM [60]. These scaffolds can be composed of biocompatible materials, such as collagen, hyaluronic acid, chitosan, or synthetic polymers (e.g., PLGA, PCL), functionalized with neurotrophic factors or adhesion molecules (e.g., laminin, fibronectin). The nanoscale topography, including fiber orientation and pore size, plays a crucial role in guiding axonal alignment, enhancing neurite extension, and supporting cell migration [60]. Electrospun nanofibers, in particular, can be aligned to direct axonal regrowth along specific paths, mimicking the organized structure of white matter tracts. To further enhance regenerative outcomes, scaffolds can be functionalized with bioactive agents (e.g., BDNF, NGF, VEGF) that promote neuronal survival, angiogenesis, and synaptic plasticity [60,80]. Additionally, incorporating conductive nanomaterials, such as graphene or carbon nanotubes, allows scaffolds to transmit electrical signals, synergistically stimulating neuronal activity and enhancing axon regeneration [61]. Recent developments have introduced multifunctional scaffolds that are capable of simultaneously clearing reactive oxygen species (ROS), modulating inflammation, and stimulating electrical activity, thereby creating a pro-regenerative microenvironment [61] (Table 3). Combining nanoscaffolds with stem cells or induced oligodendrocyte precursor cells (iOPCs) offers an even more powerful regenerative strategy. For instance, human umbilical cord mesenchymal stem cells differentiated into OPCs using TET3 overexpression have demonstrated the capacity to promote remyelination and functional recovery after SCI [77]. By embedding these cells into nanostructured matrices, it is possible to provide physical support, biochemical cues, and protection from hostile inflammatory environments, thereby enhancing cell survival, differentiation, and integration into host tissue [77]. The therapeutic efficacy of nanotechnologies for CNS injuries critically depends on their interactions with the various cellular players involved in injury and repair processes (Table 5). Understanding how nanomaterials influence microglia, astrocytes, neurons, and oligodendrocytes is essential for optimizing regenerative strategies and minimizing adverse effects. Following injury, microglia rapidly become activated and can adopt either a pro-inflammatory (M1) or pro-regenerative (M2) phenotype. Similarly, astrocytes may undergo reactive astrogliosis, leading to scar formation and the inhibition of axonal regeneration [6]. Nanomaterials can modulate these glial responses in a beneficial way. Certain nanoparticles, such as Prussian blue nanoparticles and antioxidant nanomaterials, have been shown to reduce pro-inflammatory cytokine production (e.g., TNF-α, IL-1β) and promote M2 polarization of microglia, supporting a regenerative environment [3,61]. Surface-modified nanocarriers can interact with astrocytes to limit glial scar formation and enhance the secretion of neurotrophic factors [60]. By reprogramming the inflammatory response, nanotechnologies can shift the microenvironment from a neurotoxic to a neuroprotective state. Axonal damage and demyelination are major contributors to functional loss after CNS injury. Nanostructured materials contribute to axonal repair through promoting axonal regrowth: aligned electrospun nanofibers provide topographical guidance that stimulates axon elongation along the desired trajectories [60]; enhancing remyelination: conductive scaffolds combined with stem cell therapies (e.g., iOPCs) can support oligodendrocyte maturation and efficient remyelination. Nanocarriers delivering growth factors, such as IGF-1 and PDGF-AA, or gene editing tools (e.g., CRISPR-Cas9 systems) may further potentiate axonal repair [60,77]. Recovery of CNS function also requires the restoration of synaptic networks. Nanoparticles can aid in synaptic plasticity by delivering BDNF, NT-3, or other synaptogenic molecules directly to the injury site [60,80]; modulating the intracellular signaling cascades (e.g., Ca^2+^ influx, cAMP/PKA pathways) that govern synaptic remodeling; and acting as nanoscaffolds that facilitate cell–cell contacts and synaptogenesis by mimicking the 3D nanoarchitecture of synaptic clefts. This targeted support of synaptic repair is crucial for regaining motor, sensory, and cognitive functions after TBI and SCI.

## 6. AI-Guided Personalized Nanomedicine

Personalized medicine aims to design interventions that match the individual characteristics of each patient’s injury, genetics, and molecular profile, by predicting optimal nanoparticle formulations (e.g., size, surface chemistry, cargo) based on injury type and patient-specific biomarkers [40]; optimizing dosing schedules and delivery routes using machine learning algorithms trained on preclinical and clinical datasets; and monitoring treatment responses in real time through AI-enhanced imaging and biomarker analysis. By combining nanotechnology with AI-driven personalization, it becomes possible to maximize the therapeutic efficacy while minimizing risks, moving toward a precision neuroregenerative therapy model (Table 6).

## 7. Integration with Conventional Therapies and Personalized Nanomedicine

Although nanotechnology-based approaches have shown remarkable promise in experimental models of CNS injury, their integration with established therapeutic modalities is crucial in translating these innovations into clinical practice. Nanotechnology-based approaches can be integrated synergistically with conventional treatments, such as physical rehabilitation, exercise protocols, and gene therapies, to maximize outcomes. Exercise-induced exerkines, such as neurotrophic factors, may act synergistically with nanotherapy to enhance regeneration [81]. Recent evidence also suggests that cognitive multisensory rehabilitation, combined with advanced nanoparticle therapies, may reshape the neural networks involved in body awareness and pain modulation, providing a holistic approach to functional recovery [82].

### Physical Rehabilitation

Physical rehabilitation remains a basis in the management of TBI and SCI. Nanotherapeutics can synergize with rehabilitative strategies to enhance functional recovery, as the exercise-induced secretion of exerkines (e.g., IGF-1, VEGF, BDNF) can be potentiated by nanocarriers delivering complementary neuroprotective agents; structured exercise regimens can facilitate nanoparticle distribution and enhance cellular uptake within injured neural tissues, improving therapeutic outcomes [81] (Table 7). Recent studies also suggest that combining neuromuscular stimulation, task-specific training, and nanoscaffold implantation can promote superior motor recovery compared with isolated interventions. Even integrating cognitive rehabilitation with nanotechnology-based therapies may be particularly beneficial for addressing sensory and cognitive deficits post injury. Multisensory rehabilitation programs have been shown to induce neural plasticity and the reorganization of brain networks related to body awareness and pain processing. Smart nanomaterials that are capable of modulating synaptic activity could be paired with cognitive exercises to amplify neuroplastic changes at both the structural and functional levels [82]. This integrated approach holds promise for a more holistic recovery of both motor and cognitive domains (Table 7).

## 8. Discussion

Traumatic injuries to the brain and spinal cord remain among the most formidable challenges in clinical neuroscience, largely due to the intrinsic complexity of central nervous system (CNS) repair mechanisms. While conventional therapies provide some benefit, they often fall short of achieving full functional restoration.

Previous reviews have provided valuable overviews of nanotechnology applications in CNS repair [7,8,11]. However, many of these works have primarily catalogued nanoparticle formulations or preclinical studies, with limited critical evaluation of design parameters, biocompatibility, and translational barriers. Others have focused more on the pathophysiology of SCI without establishing explicit links to nanotechnology [9,13]. The novelty of our review lies in bridging these two perspectives: we connect injury mechanisms to specific nanotechnological strategies; analyze how nanoparticles interact with cellular players, such as microglia, astrocytes, and oligodendrocytes; and emphasize integrative, multimodal approaches that combine nanotechnology with gene therapy, rehabilitation, and AI-driven personalization. This provides not only a synthesis of the current evidence but also a roadmap highlighting opportunities and challenges for clinical translation.

Nanotechnology has emerged as a transformative frontier, offering ultra-precise diagnostic tools, targeted delivery systems for therapeutic agents, and innovative scaffolding platforms capable of overcoming many of the biological barriers to regeneration. Nanomarkers, in particular, are leading the way in precision medicine by providing highly sensitive and specific tools for molecular and functional imaging. Their integration with biomedical imaging enhances diagnostic accuracy and paves the way for image-guided interventions, early detection, and personalized therapeutic strategies.

Moreover, nanomaterials can interact dynamically with key cellular players, such as microglia, astrocytes, neurons, and axons, thereby modulating the injury microenvironment and promoting neural repair. This convergence of nanotechnology and neuroscience opens new therapeutic horizons, with the potential to significantly improve outcomes, reduce systemic side effects, and bring renewed hope to patients affected by both acute and chronic neurotrauma.

However, despite these encouraging advances, several translational challenges remain to be addressed. A critical evaluation of biocompatibility, immune compatibility, and adverse effects is therefore essential to realistically assess the potential of nanoparticle-based therapies in SCI.

While nanoparticle-based delivery systems have demonstrated substantial promise in preclinical models of spinal cord injury, their translation into clinical practice remains challenged by several fundamental issues. Biocompatibility and immune compatibility are of particular concern: polymeric nanoparticles, such as PLGA, though biodegradable, may produce acidic degradation byproducts that irritate local tissue [46]; lipid nanoparticles, despite their clinical success in mRNA vaccines, are limited by immune responses, such as the accelerated blood clearance phenomenon driven by anti-PEG antibodies [54,55]; and inorganic nanoparticles (gold, iron oxide, carbon nanotubes) risk long-term accumulation, oxidative stress, and off-target organ retention [56]. Exosome-derived carriers, while highly biocompatible, suffer from production variability and scalability barriers that impede reproducibility across laboratories [57,58]. In addition, nucleic acid cargos, such as siRNA, mRNA, or CRISPR components, may activate innate immune receptors including Toll-like receptors, which could exacerbate the inflammatory milieu that is already present in injured neural tissue [59]. These limitations underscore the importance of not only optimizing particle chemistry, size, and functionalization but also systematically addressing safety, biodistribution, and clearance in clinically relevant models.

A balanced synthesis highlights both the advantages and limitations of nanotechnology-based strategies for SCI. On the one hand, nanoparticles enable targeted delivery, controlled release, multimodal imaging, and microenvironment modulation—capabilities that conventional therapies lack [67]. On the other hand, translational barriers remain substantial. Safety concerns include long-term biocompatibility, persistent accumulation of inorganic carriers [56], immune activation by surface chemistries [54,58], and variability in degradation kinetics [46]. Scalability poses another major obstacle, since reproducible large-scale manufacturing under GMP conditions is technically demanding and costly, particularly for complex hybrid or exosome-based formulations [57,58]. Research limitations also hinder progress: most preclinical studies rely on small animal cohorts, heterogeneous injury models, and relatively short follow-up times, making it difficult to predict long-term efficacy and safety in humans [59]. Addressing these issues is essential to move the field from promising experimental data to clinically validated therapies.

Looking forward, innovation in the field will likely depend on the design of “smart” nanoparticles that are responsive to the injury microenvironment (e.g., redox-, pH-, or enzyme-sensitive systems), which can selectively release therapeutic cargos in zones of oxidative stress, inflammation, or apoptosis. Hybrid strategies that integrate nanoparticles with stem cell or extracellular vesicle platforms may further enhance both the delivery specificity and regenerative potential [57]. Multimodal nanoparticles that are capable of simultaneous imaging and therapy (theranostics) provide another avenue, enabling the real-time monitoring of biodistribution and therapeutic response [67]. Importantly, advances in personalized nanomedicine, where nanoparticle formulations are tailored to the patient’s immune profile and injury stage, may help to overcome variability and improve efficacy. Finally, progress in large-scale, GMP-compliant production and in-depth toxicological profiling will be crucial for bridging the gap between experimental proof-of-concept and clinical translation. Together, these perspectives highlight both the opportunities and challenges that must be addressed for nanomedicine to achieve a meaningful impact in the treatment of spinal cord injury.

Looking ahead, the integration of nanotechnology with conventional rehabilitation strategies and AI-driven personalized approaches is poised to usher in a new era of precision neuroregeneration. Future research should prioritize the optimization of biocompatibility, a deeper understanding of long-term biodistribution, and the advancement of translational studies to ensure the successful transition of these promising technologies from bench to bedside.

## 9. Conclusions

Traumatic injuries to the central nervous system, particularly the brain and spinal cord, remain among the most formidable challenges in regenerative medicine. Nanotechnology has emerged as a transformative paradigm, offering ultra-precise diagnostics, targeted therapeutic delivery, and bioengineered scaffolds that can overcome many of the biological barriers to neural repair.

By enabling dynamic interaction with immune and neural cells, integrating with advanced imaging systems, and supporting personalized treatment through artificial intelligence, nanomedicine provides a multi-dimensional approach to neuroregeneration. By interacting at the molecular and cellular levels, nanomaterials can modulate immune responses, promote axonal regrowth and remyelination, and provide platforms for multimodal imaging and theranostic monitoring.

A key message of this review is that the therapeutic efficacy of nanomedicine depends critically on design parameters, such as particle composition, size, surface chemistry, and functionalization, which determine biodistribution, targeting specificity, and safety. While preclinical results are highly promising, several translational barriers remain, including long-term biocompatibility, immune compatibility, large-scale GMP-compliant manufacturing, and limited validation in large animal and clinical studies.

Nevertheless, the convergence of nanotechnology, neuroscience, and personalized medicine marks a decisive step toward a new era of precision neuroregeneration. The integration of smart nanomaterials with stem cell therapy, gene editing, rehabilitation, and AI-driven personalization holds the potential to significantly enhance functional recovery and improve quality of life for patients. Addressing the current challenges through interdisciplinary collaboration will be essential to move from experimental innovation to clinically validated therapies, ultimately bringing renewed hope to individuals affected by both acute and chronic neurotrauma.

## Figures and Tables

**Figure 1 biomedicines-13-02176-f001:**
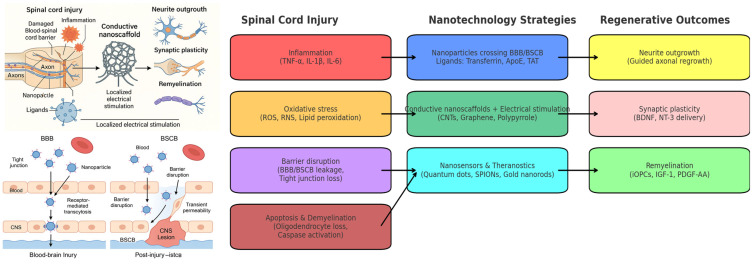
Schematic overview of the interplay between spinal cord injury (SCI) pathophysiology, nanotechnology-based therapeutic strategies, and regenerative outcomes. SCI initiates inflammatory cascades (TNF-α, IL-1β, IL-6), oxidative stress (ROS/RNS accumulation, lipid peroxidation), blood–brain/–spinal cord barrier (BBB/BSCB) disruption (tight junction loss), and apoptosis with demyelination (oligodendrocyte loss, caspase activation). To counteract these events, nanotechnology strategies exploit receptor-mediated transport across the BBB/BSCB (ligands such as transferrin, ApoE, TAT), conductive nanoscaffolds combined with electrical stimulation (CNTs, graphene, polypyrrole), and nanosensors/theranostic systems (quantum dots, SPIONs, gold nanorods) for real-time monitoring. These approaches contribute to functional recovery by promoting neurite outgrowth, enhancing synaptic plasticity (e.g., via BDNF and NT-3 delivery), and supporting remyelination (e.g., via iOPCs, IGF-1, PDGF-AA). (Schematic diagrams are designed by ChatGPT, OpenAI (version: GPT-5), based on the authors’ scientific input and conceptual guidance).

**Table 1 biomedicines-13-02176-t001:** Recent advances in nanotechnology-enabled diagnostic methodologies for CNS injury.

Nanomarker Type	Functionalization/Target	Imaging Modality	Key Advantages	Potential Applications	Examples/ References
SPIONs (Superparamagnetic Iron Oxide Nanoparticles)	Antibodies, peptides, small molecules targeting inflammation or apoptosis	MRI	High contrast, biocompatibility, long circulation, tissue-specific accumulation	Non-invasive monitoring of neuroinflammation; lesion tracking	[11,41]
Gold nanoparticles/nanorods	Surface conjugation with peptides or antibodies	Photoacoustic imaging	Strong optical absorption, deep tissue penetration, vascular/tumor structure visualization	Vascular imaging; monitoring of oxidative stress and tissue hypoxia	[42,43]
Quantum dots (QDs)	Functionalized with ligands for neuronal or glial markers	NIR fluorescence imaging	Bright emission, tunable wavelength, minimal background noise	Intraoperative surgical guidance; neuronal apoptosis detection	[43]
Carbon nanotubes	Functionalization with injury-specific ligands	Photoacoustic imaging	High optical absorption, acoustic signal conversion, hybrid optical ultrasound	Mapping vascular and structural alterations; oxidative stress detection	[42]
Radiolabeled nanoparticles (e.g., 18F, 68Ga conjugates)	PEGylation, peptides for receptor targeting	PET	Real-time metabolic tracking, high sensitivity, complements anatomical MRI	Functional neuroimaging; early detection of metabolic dysfunction after SCI	[41]
Multimodal nanoparticles (SPIONs + QDs, gold + NIR dyes)	Dual/multiple surface ligands	MRI + NIRF/ PET + PA	Combined structural + molecular information, intraoperative guidance	Pre-operative lesion mapping with MRI; real-time intraoperative NIRF guidance	[40]
Theranostic nanoparticles	Drug-loaded plus targeting ligands	MRI/ NIRF/ PET	Simultaneous therapy + diagnosis, dynamic monitoring of response	Monitoring treatment efficacy; personalized medicine approaches	[40,41]

**Table 2 biomedicines-13-02176-t002:** Comparative overview of the nanoparticle platforms for therapeutic cargo delivery in SCI.

Cargo Type	Optimal NP Design	Size/Shape Rationale	Targeting Strategies	Advantages	Limitations/Adverse Effects
Growth factors (BDNF, IGF)	Liposomes, PLGA NPs	>150 nm sustain release, protein stability	PEGylation, antibody conjugation	Protect proteins, controlled release	Burst release, acidic byproducts (PLGA irritation)
siRNA/mRNA	Lipid nanoparticles, exosomes, cationic polymers	<100 nm efficient endocytosis and escape	RVG peptide (neurons), mannose (microglia), transferrin (BBB)	Efficient transfection, systemic delivery	Immune activation (TLR), PEG immune reactions
CRISPR/Cas components	Hybrid lipid–polymer NPs, gold NPs	<120 nm facilitate nuclear delivery	Nuclear localization peptides, antibody targeting	Genome editing, long-term correction	Off-target effects, immune sensing of Cas proteins
Small molecules/ antioxidants	Inorganic NPs (Au, SPIONs), polymeric micelles	Variable; spheres more stable	Often passive or minimal	ROS scavenging, magnetic/optical guidance	Accumulation, oxidative stress, organ retention

**Table 6 biomedicines-13-02176-t006:** Summary of the key types of nanoparticles being investigated for the treatment of SCI, their therapeutic cargos, main advantages and limitations, and current clinical development status.

Nanoparticle Type	Therapeutic Application	Clinical Status	Representative Outcomes	Clinical Trial ID (NCT#)
Liposomes	Delivery of corticosteroids, neuroprotective agents	Evaluated in neurological disorders	Improved drug stability and bioavailability	
Iron oxide nanoparticles (Ferumoxytol)	Imaging and theranostics	Approved for anemia; tested in CNS imaging	Enhanced MRI contrast at lesion sites	
Exosomes/extracellular vesicles	Delivery of RNA, proteins; regenerative therapies	Phase I/II in oncology and neurodegenerative diseases	Good safety profile; immune compatibility	NCT03608631
Polymeric nanoparticles (PLGA, PEG-PLA)	Growth factors, nucleic acids (preclinical SCI, oncology)	Some formulations in early phase clinical trials	Sustained release; neuroprotection	NCT04314895

**Table 7 biomedicines-13-02176-t007:** Summary of the emerging applications of nanotechnology designed to enhance rehabilitation outcomes in spinal cord injury (SCI). Each row highlights a distinct area of application, describing the mechanism, functional impact, and development stage of the technology.

Application Area	Description	Rehabilitative Benefit	Development Status
Neurotrophic Factor Delivery	Use of nanoparticles to deliver agents like BDNF, NGF, or IGF-1 to enhance plasticity during motor rehabilitation.	Amplifies the effect of activity-based therapies by promoting synaptic and axonal plasticity.	Preclinical
Bioelectronic Interfaces	Integration of conductive nanomaterials into scaffolds or implants to restore electrical signaling and support neuronal reactivation.	Enables the functional reactivation of spinal circuits and synergy with FES or robotic training.	Preclinical to early prototyping
Nanosensors for Monitoring	Implantable or wearable nanosensors to monitor inflammation, neural activity, or metabolic markers during therapy.	Personalizes rehabilitation intensity and timing based on real-time physiological data.	Emerging technology
Gene Modulation via Nanocarriers	Nanoparticles carrying siRNA or miRNA to modulate genes involved in inhibitory signaling or regeneration during rehabilitation phases.	Maximizes the molecular environment’s responsiveness to training by modulating key signaling pathways.	Preclinical studies in animal models

## Data Availability

No new data were created or analyzed in this study. Data sharing is not applicable to this article.

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
