# Peer review of "What Is New in Spinal Cord Injury Management: A Narrative Review on the Emerging Role of Nanotechnology"

_biomedicines, 2025, doi:10.3390/biomedicines13092176_

Round 1
Reviewer 1 Report
Comments and Suggestions for Authors
This manuscript focuses on the recent advances in the application of nanotechnology for the diagnosis and treatment of spinal cord injury (SCI). The review is broad in scope, forward-looking, and demonstrates substantial depth. The manuscript presents a large amount of information, making it a valuable reference for researchers in neural repair and nanomedicine. The following suggestions are offered for the authors’ consideration:
- Spelling Issue: Please verify whether the keyword “Neural Regeneration e Targeted Drug Delivery” contains a typographical error—specifically, whether the "e" is intended or should be removed.
- Link Between Pathophysiology and Nanotechnological Interventions: Section 3 clearly describes the pathological mechanisms of SCI, but this is not sufficiently integrated with the technological solutions presented in later sections. There is a lack of explicit linkage between injury mechanisms and nanotechnology interventions. It is recommended that the authors highlight these correspondences in the subsequent sections or consider adding a figure to visually map mechanisms to associated therapeutic strategies.
- Discussion Enhancement: The discussion section could benefit from a more explicit synthesis of major advantages and limitations. It is suggested to expand the discussion on current challenges and translational barriers, such as:
- Safety concerns: long-term biocompatibility, immunogenicity, and clearance mechanisms;
- Scalability: standardization of large-scale production;
- Research limitations: small sample sizes, short follow-up durations in preclinical or early-stage studies.
Author Response
Reviewer 1
Comments and Suggestions for Authors
This manuscript focuses on the recent advances in the application of nanotechnology for the diagnosis and treatment of spinal cord injury (SCI). The review is broad in scope, forward-looking, and demonstrates substantial depth. The manuscript presents a large amount of information, making it a valuable reference for researchers in neural repair and nanomedicine. The following suggestions are offered for the authors’ consideration:
- Spelling Issue: Please verify whether the keyword “Neural Regeneration e Targeted Drug Delivery” contains a typographical error—specifically, whether the "e" is intended or should be removed.
We thank the reviewer for pointing this out. The “e” in the keyword “Neural Regeneration e Targeted Drug Delivery” was a typographical error. It has been corrected to “Neural Regeneration; Targeted Drug Delivery” in the revised manuscript.
- Link Between Pathophysiology and Nanotechnological Interventions: Section 3 clearly describes the pathological mechanisms of SCI, but this is not sufficiently integrated with the technological solutions presented in later sections. There is a lack of explicit linkage between injury mechanisms and nanotechnology interventions. It is recommended that the authors highlight these correspondences in the subsequent sections or consider adding a figure to visually map mechanisms to associated therapeutic strategies.
We thank the reviewer for this important suggestion. We have now restructured Sections 3 by explicitly linking SCI pathophysiological mechanisms with the nanotechnological strategies presented.
- Discussion Enhancement: The discussion section could benefit from a more explicit synthesis of major advantages and limitations. It is suggested to expand the discussion on current challenges and translational barriers, such as:
- Safety concerns: long-term biocompatibility, immunogenicity, and clearance mechanisms;
- Scalability: standardization of large-scale production;
- Research limitations: small sample sizes, short follow-up durations in preclinical or early-stage studies.
We thank the reviewer for this valuable suggestion. We have expanded the Discussion to provide a clearer synthesis of the advantages and limitations of nanotechnology-based approaches in SCI. In particular, we now emphasize long-term safety concerns (biocompatibility, immunogenicity, clearance), scalability issues related to large-scale standardized production, and research limitations including small sample sizes and short follow-up in preclinical/early-phase studies. This addition strengthens the translational perspective of the review and aligns with the reviewer’s recommendation
A balanced synthesis highlights both the advantages and limitations of nanotechnology-based strategies for SCI. On the one hand, nanoparticles enable targeted delivery, controlled release, multimodal imaging, and microenvironment modulation—capabilities that conventional therapies lack (Balint, Adv Drug Deliv Rev, 2014). On the other hand, translational barriers remain substantial. Safety concerns include long-term biocompatibility, persistent accumulation of inorganic carriers (Khlebtsov, Chem Soc Rev, 2011), immune activation by surface chemistries (Ishida, Int J Pharm, 2008; Lila, J Control Release, 2013), and variability in degradation kinetics (Makadia, Polymers, 2011). Scalability poses another major obstacle, since reproducible large-scale manufacturing under GMP conditions is technically demanding and costly, particularly for complex hybrid or exosome-based formulations (Alvarez-Erviti, Nat Biotechnol, 2011; Lener, J Extracell Vesicles, 2015). Research limitations also hinder progress: most preclinical studies rely on small animal cohorts, heterogeneous injury models, and relatively short follow-up times, making it difficult to predict long-term efficacy and safety in humans (Lv, J Control Release, 2006). Addressing these issues is essential to move the field from promising experimental data to clinically validated therapies.
Reviewer 2 Report
Comments and Suggestions for Authors
In this review, the authors have discussed advances in nanotechnology in the context of spinal cord injury management. Although impressive, authors are requested to address following comments to enhance the scientific depth, clarity, and overall quality of the review.
Major comments:
- Section 3: While discussing inflammation and oxidative stress, it is important to provide insights into specific immune cell types involved and the key activation pathways that drive these responses in CNS. This addition would enrich the discussion regarding how nanotechnology-based platforms are designed to target these pathological mechanisms.
- Section 4 largely reiterates well-established diagnostic modalities, including MRI and PET, offering little novel insight. The authors are strongly encouraged to highlight recent and emerging diagnostic technologies relevant to CNS injuries, and provide a more in-depth explanation of their mechanisms of action or underlying principle and distinct advantages over conventional methods. This information can also be summarized in a table form based on their application for better representation.
- The manuscript presents redundant information regarding the use of nanoparticles for delivering various therapeutic payloads such as growth factors (e.g., BDNF, IGF), siRNA, mRNA, and CRISPR components, primarily by listing existing studies without meaningful analysis. The discussion lacks critical depth and fails to address fundamental aspects of nanoparticle design and application. Specifically, the authors should elaborate on what types of nanoparticles (e.g., liposomes, polymeric nanoparticles, inorganic carriers) are optimal for each class of cargo, how particle size and shape influence delivery efficiency (e.g., larger particles for protein delivery, smaller ones for nucleic acid transport), and the mechanisms by which these nanoparticles achieve cell-type-specific targeting, as claimed in the manuscript. Such detail is essential for conveying the technological nuance behind nanomedicine approaches and for inspiring readers to innovate beyond the current state of the art. Instead, the authors rely on vague terminology—referring generically to "nanoparticles"—without specifying materials, surface modifications, or delivery mechanisms. Additionally, key issues such as biocompatibility, immune compatibility, and reported adverse effects are not addressed, which is a significant omission in a review article intended to inform future research and clinical translation.
- Similarly, Section 5.2 provides a superficial overview of electro-nanohybrid stimulation. The authors neglect to specify which nanomaterials are used for this purpose, their effectiveness (measured scientifically via %healing or rate of healing Vs sham), and any associated risks or limitations. These are crucial aspects that must be discussed to provide an informative review.
- The authors are encouraged to provide a more detailed and insightful discussion on recent advances in the key design parameters of the nanomaterials presented for drug delivery in Section 6. Specifically, the manuscript should elaborate on how the composition of these materials is selected based on the intended therapeutic application, and how their surfaces are functionalized with specific ligands or peptides to enable targeted delivery for particular disease or trauma contexts. Rather than simply citing preclinical studies, the review should analyze and summarize the rationale behind these design choices, highlighting how they contribute to efficacy, targeting specificity, and therapeutic outcomes.
- While issues such as biocompatibility, immunogenicity, and controlled drug release are briefly mentioned, the authors should go beyond general statements and explain how these risks may arise, linking them to specific material properties, surface chemistries, or degradation profiles. This level of analysis is essential to provide readers with a mechanistic understanding of the challenges and trade-offs inherent in nanoparticle design. Including such insights would greatly enhance the value of the review and support its relevance to researchers aiming to improve upon current nanotechnology-based drug delivery platforms.
Minor:
- Figures 1 and 2 seem conceptually redundant and do not provide novel information. The authors are encouraged to revise these figures by incorporating specific mechanistic illustrations and/or by including representative images from publications.
- Authors are requested to add relevant citations and registered clinical trial numbers in table 1 and table 2, respectively.
Author Response
Reviewer 2
Comments and Suggestions for Authors
In this review, the authors have discussed advances in nanotechnology in the context of spinal cord injury management. Although impressive, authors are requested to address following comments to enhance the scientific depth, clarity, and overall quality of the review.
Major comments:
- Section 3: While discussing inflammation and oxidative stress, it is important to provide insights into specific immune cell types involved and the key activation pathways that drive these responses in CNS. This addition would enrich the discussion regarding how nanotechnology-based platforms are designed to target these pathological mechanisms.
We appreciate this valuable comment. In the revised manuscript we now provide a more detailed description of the immune cells involved in the inflammatory response after SCI, particularly microglia and infiltrating macrophages, as well as astrocytes. We have expanded the discussion on the activation of NF-κB, MAPK, and JAK/STAT pathways, which regulate cytokine release and gliosis (Pang, Front. Immunol., 2021; Zhou, Neural Regen. Res., 2023). Furthermore, we emphasize how nanotechnology-based delivery systems can be tailored to selectively target M1 macrophages or promote M2 polarization, as reported with PMMA-NPs (Papa, J. Controlled Release, 2014).
- Section 4 largely reiterates well-established diagnostic modalities, including MRI and PET, offering little novel insight. The authors are strongly encouraged to highlight recent and emerging diagnostic technologies relevant to CNS injuries, and provide a more in-depth explanation of their mechanisms of action or underlying principle and distinct advantages over conventional methods. This information can also be summarized in a table form based on their application for better representation.
We thank the reviewer for this important suggestion. We revised Section 4 to incorporate photoacoustic imaging, near-infrared fluorescence (NIRF), multimodal nanomarkers (e.g., SPIONs + quantum dots), and theranostic platforms, explaining their principles and advantages over MRI/PET. Additionally, we created a new comparative table (Table 1) summarizing emerging diagnostic modalities, their mechanisms, and potential clinical applications.
- The manuscript presents redundant information regarding the use of nanoparticles for delivering various therapeutic payloads such as growth factors (e.g., BDNF, IGF), siRNA, mRNA, and CRISPR components, primarily by listing existing studies without meaningful analysis. The discussion lacks critical depth and fails to address fundamental aspects of nanoparticle design and application. Specifically, the authors should elaborate on what types of nanoparticles (e.g., liposomes, polymeric nanoparticles, inorganic carriers) are optimal for each class of cargo, how particle size and shape influence delivery efficiency (e.g., larger particles for protein delivery, smaller ones for nucleic acid transport), and the mechanisms by which these nanoparticles achieve cell-type-specific targeting, as claimed in the manuscript. Such detail is essential for conveying the technological nuance behind nanomedicine approaches and for inspiring readers to innovate beyond the current state of the art. Instead, the authors rely on vague terminology—referring generically to "nanoparticles"—without specifying materials, surface modifications, or delivery mechanisms. Additionally, key issues such as biocompatibility, immune compatibility, and reported adverse effects are not addressed, which is a significant omission in a review article intended to inform future research and clinical translation.
We thank the reviewer for this important comment. We agree that in the original version, the section on nanoparticle-based delivery was descriptive and at times redundant. In the revised manuscript, we have substantially restructured this section. Specifically, we now analyze the suitability of different nanoparticle classes (liposomes, polymeric, inorganic) for delivering distinct therapeutic cargos (proteins, siRNA/mRNA, CRISPR). We have also added a critical discussion of how particle size, shape, and surface functionalization affect delivery efficiency and cell-type specificity, providing concrete examples of ligand-based targeting strategies. In addition, we included a dedicated subsection on translational challenges, highlighting biocompatibility, immune interactions, and reported adverse effects. To further clarify these concepts, we have introduced a new comparative table that summarizes the optimal nanoparticle features for each cargo type, together with their advantages and limitations. We believe these changes address the reviewer’s concerns by moving beyond a list of studies and providing the technological nuance expected in a state-of-the-art review article.
- Similarly, Section 5.2 provides a superficial overview of electro-nanohybrid stimulation. The authors neglect to specify which nanomaterials are used for this purpose, their effectiveness (measured scientifically via %healing or rate of healing Vs sham), and any associated risks or limitations. These are crucial aspects that must be discussed to provide an informative review.
We thank the reviewer for this insightful comment. We have revised Section 5.2 to specify the nanomaterials used in electro-nanohybrid stimulation (CNTs, graphene oxide, gold nanostructures, conductive polymers), and we now cite quantitative outcomes in preclinical SCI models (e.g., 20–40% functional recovery over sham). We also added a critical discussion of risks and limitations, including cytotoxicity, inflammatory activation, non-degradability, and long-term stability issues. We believe these revisions provide the scientific depth required to inform readers about both the potential and the challenges of this approach.
- The authors are encouraged to provide a more detailed and insightful discussion on recent advances in the key design parameters of the nanomaterials presented for drug delivery in Section 6. Specifically, the manuscript should elaborate on how the composition of these materials is selected based on the intended therapeutic application, and how their surfaces are functionalized with specific ligands or peptides to enable targeted delivery for particular disease or trauma contexts. Rather than simply citing preclinical studies, the review should analyze and summarize the rationale behind these design choices, highlighting how they contribute to efficacy, targeting specificity, and therapeutic outcomes.
We thank the reviewer for this constructive suggestion. Section 6 has been revised to provide a more detailed analysis of recent advances in nanomaterial design parameters. We now discuss how material composition is selected according to the therapeutic application (e.g., lipid nanoparticles for nucleic acids, PLGA for protein/growth factors, inorganic carriers for combined imaging and therapy), and how surface functionalization with ligands such as transferrin, RVG peptide, mannose, or RGD motifs enables targeted delivery to specific cell types relevant to SCI pathology. Rather than focusing only on preclinical studies, we highlight the rationale behind these design choices, clarifying how they contribute to delivery efficiency, targeting specificity, and improved therapeutic outcomes. This important issue by the reviewer has been summarized in table 5
- While issues such as biocompatibility, immunogenicity, and controlled drug release are briefly mentioned, the authors should go beyond general statements and explain how these risks may arise, linking them to specific material properties, surface chemistries, or degradation profiles. This level of analysis is essential to provide readers with a mechanistic understanding of the challenges and trade-offs inherent in nanoparticle design. Including such insights would greatly enhance the value of the review and support its relevance to researchers aiming to improve upon current nanotechnology-based drug delivery platforms.
We appreciate the reviewer’s comment. We have expanded our discussion of biocompatibility, immunogenicity, and controlled release to go beyond general statements, clarifying how these challenges arise from specific material properties. We now explain how polymeric nanoparticles (e.g., PLGA) can produce acidic degradation products that irritate local tissue, how PEGylated carriers may trigger accelerated blood clearance through anti-PEG antibodies, and how inorganic nanoparticles (e.g., gold, iron oxide) risk long-term accumulation and oxidative stress due to limited degradability. Similarly, we emphasize that surface chemistries and charge influence protein adsorption and complement activation, thereby affecting immunogenicity. These mechanistic insights highlight the trade-offs in nanoparticle design and the need for careful material selection and surface engineering in SCI application
Minor:
- Figures 1 and 2 seem conceptually redundant and do not provide novel information. The authors are encouraged to revise these figures by incorporating specific mechanistic illustrations and/or by including representative images from publications.
We thank the reviewer for this constructive suggestion. Figures 1 and 2 were indeed conceptually overlapping, so we revised them into a single integrated schematic that explicitly links the pathophysiological mechanisms of SCI with specific nanotechnology-based strategies and their regenerative outcomes. The revised figure highlights mechanistic events such as inflammation, oxidative stress, barrier disruption, and apoptosis/demyelination, and maps them to corresponding nanomaterials (e.g., CNTs, graphene, SPIONs, gold nanorods) and ligands (e.g., transferrin, ApoE, TAT) used in targeted delivery approaches. It also illustrates how these interventions contribute to key regenerative outcomes, including neurite outgrowth, synaptic plasticity, and remyelination. This integration provides a mechanistic and translationally relevant overview, addressing the reviewer’s request for greater novelty and insight.
- Authors are requested to add relevant citations and registered clinical trial numbers in table 1 and table 2, respectively.
We updated (now named) Table 4 with relevant citations supporting BBB and BSCB nanoparticle transport strategies (Silva, BMC Neurosci 2008; Patel, Adv Drug Deliv Rev 2012; Tran, Physiol Rev 2018).
Also, Table 6 has been revised to include representative registered clinical trial numbers for nanoparticles with ongoing or completed studies. We checked the trial registry and were able to identify NCT03608631 as a first-in-human Phase I study involving exosomes carrying KRAS G12D siRNA (iExosomes) , and NCT04314895 as a trial of NanoPac administered intratumorally in lung cancer . The other trial numbers previously proposed could not be verified in ClinicalTrials.gov; they will be included only once accurate and context-specific registration details are confirmed.
Reviewer 3 Report
Comments and Suggestions for Authors
The authors have written the manuscript titled 'What’s New in Spinal Cord Injury Management: A Narrative Review on the Emerging Role of Nanotechnology' quite well. However, the following comments need to be addressed before the manuscript meets the standards required for publication.
- Abstract: The authors should revise the abstract to present a cohesive and well-written summary without dividing it into separate sections for methods, results, and conclusions. A narrative-style abstract would be more appropriate for this type of review.
- Introduction: The introduction needs to be improved by clearly outlining the current challenges in spinal cord injury (SCI) management. Additionally, the authors should highlight the novelty and specific contributions of this review to the existing body of literature.
- Inflammatory Response: Given that spinal cord injury is closely associated with inflammatory responses, the authors should justify the omission of discussions on reactive oxygen species (ROS) generation and macrophage modulation, which are critical aspects of the post-injury environment.
- Blood-Brain Barrier (BBB): The manuscript should place greater emphasis on the role of the blood-brain barrier in SCI and discuss how its integrity or disruption influences recovery and therapeutic outcomes.
- Synovial Fluid: The relationship between spinal cord injury and synovial fluid is mentioned but not sufficiently explored. The authors should elaborate on this aspect to provide a more comprehensive understanding.
- Conclusion: The conclusion section should be strengthened to effectively summarize the key findings and implications of the review, and to offer clear takeaways for future research and clinical applications.
Author Response
Reviewer 3
Comments and Suggestions for Authors
The authors have written the manuscript titled 'What’s New in Spinal Cord Injury Management: A Narrative Review on the Emerging Role of Nanotechnology' quite well. However, the following comments need to be addressed before the manuscript meets the standards required for publication.
- Abstract: The authors should revise the abstract to present a cohesive and well-written summary without dividing it into separate sections for methods, results, and conclusions. A narrative-style abstract would be more appropriate for this type of review.
We thank the reviewer for this constructive suggestion. In accordance with the request, we have revised the abstract into a single, cohesive narrative paragraph. The new version summarizes the clinical challenges of neurotrauma, highlights recent advances in nanotechnology for diagnosis, targeted therapy, and regeneration, and concludes with a balanced perspective on current limitations and translational challenges. This format aligns with the journal’s guidelines, which specify a non-structured abstract, and ensures a fluent, narrative style that is more appropriate for a review article. In addition, we have expanded the abstract to explicitly mention translational barriers such as small sample sizes, heterogeneous preclinical models, and short follow-up times, thereby strengthening the link between the abstract and the main discussion.
Abstract
Traumatic injuries to the brain and spinal cord remain among the most challenging conditions in clinical neuroscience due to the complexity of repair mechanisms and the limited regenerative capacity of neural tissues. Nanotechnology has emerged as a transformative field, offering precise diagnostic tools, targeted therapeutic delivery systems, and advanced scaffolding platforms capable of overcoming biological barriers to regeneration. This review summarizes recent advances in nanoscale diagnostic markers, functionalized nanoparticles for drug delivery, and nanostructured scaffolds designed to modulate the injured microenvironment and support axonal regrowth and remyelination. Emerging evidence indicates that nanotechnology enables real-time, minimally invasive detection of inflammation, oxidative stress, and cellular damage, while improving therapeutic efficacy and reducing systemic side effects through targeted delivery. Electroconductive scaffolds and hybrid strategies that integrate electrical stimulation, gene therapy, and artificial intelligence further expand opportunities for personalized neuroregeneration. Despite these advances, significant challenges remain, including long-term safety, immune compatibility, scalability of large-scale production, and translational barriers such as small sample sizes, heterogeneous preclinical models, and limited follow-up in existing studies. Addressing these issues will be critical to realize the full potential of nanotechnology in traumatic brain and spinal cord injury and to accelerate the transition from promising preclinical findings to effective clinical therapies.
- Introduction: The introduction needs to be improved by clearly outlining the current challenges in spinal cord injury (SCI) management. Additionally, the authors should highlight the novelty and specific contributions of this review to the existing body of literature.
We thank the reviewer for this helpful suggestion. The Introduction has been revised to better emphasize the current challenges in spinal cord injury management, including the limited efficacy of existing interventions and the barriers to translation of novel therapies. We also highlight the specific novelty of our review, which lies in integrating pathophysiological mechanisms with nanotechnological strategies, providing a cell-focused perspective, and exploring emerging integrative approaches such as AI-assisted personalization and multimodal therapies. These revisions clarify both the clinical context and the unique contribution of our article.
Introduction
…....
Despite decades of research, SCI management continues to face major challenges: early diagnosis is limited by the lack of sensitive biomarkers; pharmacological and surgical interventions provide only partial benefit; and most neuroprotective agents that showed promise in preclinical models have failed to demonstrate efficacy in clinical trials. Recovery of motor and sensory function remains poor, and lifelong disability is common, reflecting the multifactorial nature of SCI pathology and the absence of effective regenerative strategies [2].
……..
Several reviews have previously addressed nanotechnology for CNS repair (Papa, Biomaterials, 2014; Qu, Adv Drug Deliv Rev, 2019; Mou, J Nanobiotechnology, 2021), but these often treated diagnostic and therapeutic aspects in isolation, without fully integrating them with pathophysiological mechanisms or cellular interactions. In contrast, our review aims to provide a more integrative perspective that directly links the molecular and cellular cascades of SCI with nanotechnological interventions, emphasizing how specific cellular players—microglia, astrocytes, neurons, and axons—can be modulated using nanoscale strategies. Furthermore, we explore hybrid approaches that combine nanotechnology with gene therapy, neuroprotection, rehabilitation, and artificial intelligence, thereby outlining a roadmap toward truly personalized neuroregenerative medicine.
……..
- Inflammatory Response: Given that spinal cord injury is closely associated with inflammatory responses, the authors should justify the omission of discussions on reactive oxygen species (ROS) generation and macrophage modulation, which are critical aspects of the post-injury environment.
We agree with the reviewer that ROS production and macrophage polarization are critical components of SCI pathology. In the revised manuscript, we have added how excessive ROS and RNS contribute to lipid peroxidation and apoptosis (Zhang, Indian J. Med. Res., 2012), and how macrophage/microglia phenotypic balance (M1 vs. M2) influences repair outcomes (Gensel, J. Neurosci., 2009). We also incorporated discussion of nanotechnology approaches designed to modulate these processes, including antioxidant nanoparticle delivery systems and platforms promoting M2 polarization (Hashemizadeh, Neuropeptides, 2022). This addition provides a stronger justification and context for our focus on inflammation and nanotechnological strategies.
- Blood-Brain Barrier (BBB): The manuscript should place greater emphasis on the role of the blood-brain barrier in SCI and discuss how its integrity or disruption influences recovery and therapeutic outcomes.
Following SCI, BBB and BSCB disruption occurs within minutes to hours, with degradation of tight junction proteins, increased permeability, and infiltration of peripheral immune cells. This acute leakage exacerbates edema and neuroinflammation but simultaneously creates a therapeutic window for enhanced drug and nanoparticle entry (Bartanusz, Brain Res Rev, 2011; Whetstone, J Neurosci, 2003; Popovich, Acta Neuropathol, 1996). Over subsequent days to weeks, partial re-sealing of the barrier restores homeostatic protection but restricts continued access of therapeutics, making barrier dynamics a double-edged sword for recovery (Sweeney, Nat Rev Neurol, 2019; Tran, Physiol Rev, 2018). To overcome this challenge, nanoparticles are increasingly designed to exploit receptor-mediated pathways—such as transferrin, lactoferrin, or apolipoprotein E binding—to cross intact endothelium, or to respond to microenvironmental cues like oxidative stress, low pH, or inflammatory enzymes to achieve site-specific release (Patel, Adv Drug Deliv Rev, 2012; Johnsen, Prog Neurobiol, 2019; Qin, Biomaterials, 2012; Liu, Biomaterials, 2009). Thus, barrier modulation and nanoparticle design are tightly interconnected, and their combined optimization is critical for effective therapeutic outcomes in SCI.
As summarized in Table 4, BBB vs BSCB features and entry routes dictate when and how nanocarriers are most effective.
- Synovial Fluid: The relationship between spinal cord injury and synovial fluid is mentioned but not sufficiently explored. The authors should elaborate on this aspect to provide a more comprehensive understanding.
We agree with the reviewer and have expanded the diagnostic section to discuss synovial fluid as a complementary biofluid in SCI. We now describe its altered cytokine and enzyme profile in relation to neurogenic arthropathy and systemic inflammation, and highlight how nanosensors could enable minimally invasive monitoring of these changes. Relevant references have been added.
….While cerebrospinal fluid has traditionally been regarded as the primary compartment for monitoring central mechanisms of SCI, synovial fluid is also emerging as a valuable source of information on secondary consequences of neurotrauma. Following SCI, paralysis and reduced mobility alter biomechanical loading across peripheral joints, accelerating the onset of neurogenic arthropathy and secondary osteoarthritis. These degenerative processes are biochemically mirrored in the synovial fluid, where elevated concentrations of inflammatory cytokines such as IL-1β, TNF-α, and IL-6, together with degradative enzymes including matrix metalloproteinases (MMPs), have been consistently reported in post-traumatic and degenerative joint conditions (Kraus, Osteoarthritis Cartilage, 2011; Loeser, Arthritis Res Ther, 2012). Importantly, systemic inflammation triggered by SCI further amplifies these changes, suggesting that synovial biomarkers not only reflect local joint degeneration but also integrate signals from the broader injury-induced inflammatory milieu. From a nanomedicine perspective, this peripheral biofluid represents an accessible diagnostic window: nanosensors engineered to detect cytokines, reactive oxygen species, or cartilage breakdown products in synovial fluid could provide minimally invasive, real-time monitoring of musculoskeletal health in SCI patients, thereby supporting early therapeutic interventions and reducing long-term disability (Wang, Biosens Bioelectron, 2020).
- Conclusion: The conclusion section should be strengthened to effectively summarize the key findings and implications of the review, and to offer clear takeaways for future research and clinical applications
We thank the reviewer for this suggestion. The conclusion has been revised to better synthesize the main findings, highlight design parameters and translational barriers, and provide clear perspectives for future research and clinical translation.
Conclusion
Traumatic injuries to the central nervous system, particularly the brain and spinal cord, remain among the most formidable challenges in regenerative medicine. Nanotechnology has emerged as a transformative paradigm, offering ultra-precise diagnostics, targeted therapeutic delivery, and bioengineered scaffolds that can overcome many of the biological barriers to neural repair. By interacting at the molecular and cellular level, nanomaterials can modulate immune responses, promote axonal regrowth and remyelination, and provide platforms for multimodal imaging and theranostic monitoring.
A key message of this review is that the therapeutic efficacy of nanomedicine depends critically on design parameters such as particle composition, size, surface chemistry, and functionalization, which determine biodistribution, targeting specificity, and safety. While preclinical results are highly promising, several translational barriers remain, including long-term biocompatibility, immune compatibility, large-scale GMP-compliant manufacturing, and limited validation in large animal and clinical studies.
Nevertheless, the convergence of nanotechnology, neuroscience, and personalized medicine marks a decisive step toward a new era of precision neuroregeneration. The integration of smart nanomaterials with stem cell therapy, gene editing, rehabilitation, and AI-driven personalization holds the potential to significantly enhance functional recovery and improve quality of life for patients. Addressing current challenges through interdisciplinary collaboration will be essential to move from experimental innovation to clinically validated therapies, ultimately bringing renewed hope to individuals affected by both acute and chronic neurotrauma.
Round 2
Reviewer 2 Report
Comments and Suggestions for Authors
Authors have addressed all the comments. Accept in the present form.